# MicroRNA Landscape in Hepatocellular Carcinoma: Metabolic Re-Wiring, Predictive and Diagnostic Biomarkers, and Emerging Therapeutic Targets

**DOI:** 10.3390/biomedicines13092243

**Published:** 2025-09-11

**Authors:** Dimitris Liapopoulos, Panagiotis Sarantis, Theodora Biniari, Thaleia-Eleftheria Bousou, Eleni-Myrto Trifylli, Ioanna A. Anastasiou, Stefania Kokkali, Dimitra Korakaki, Spyridon Pantzios, Evangelos Koustas, Ioannis Elefsiniotis, Michalis V. Karamouzis

**Affiliations:** 1Biopathological Laboratory, General and Oncology Hospital “Agioi Anargyroi”, National and Kapodistrian University of Athens, Timiou Stavrou 14, 145 64 Kifisia, Greece; dimitrisliapop@gmail.com (D.L.); biniaridora@yahoo.com (T.B.); 2Biopathological Laboratory, Athens Medical Group, Psychiko Clinic, Antersen 1, 115 25 Psychiko, Greece; 3University Pathology Clinic, General and Oncology Hospital “Agioi Anargyroi”, National and Kapodistrian University of Athens, Timiou Stavrou 14, 145 64 Kifisia, Greece; panayotissarantis@gmail.com (P.S.); thaliaelb@yahoo.com (T.-E.B.); ielefs@nurs.uoa.gr (I.E.); mkaramouz@med.uoa.gr (M.V.K.); 4Institute of Molecular Medicine and Biomedical Research, 115 27 Athens, Greece; trif.lena@gmail.com (E.-M.T.); dimkorakaki@gmail.com (D.K.); 5Diabetes Center, First Department of Propaedeutic Internal Medicine, Medical School, National and Kapodistrian University of Athens, Laiko General Hospital, 115 27 Athens, Greece; anastasiouiwanna@gmail.com; 6Department of Pharmacology, Medical School, National and Kapodistrian University of Athens, 115 27 Athens, Greece; 7Oncology Unit, Second Department of Medicine, University of Athens, Hippocratio General Hospital of Athens, 115 27 Athens, Greece; stefaniakokkali8@gmail.com; 8Hepatogastroenterology Unit, Academic Department of Internal Medicine, General and Oncology Hospital “Agioi Anargyroi”, National and Kapodistrian University of Athens, 145 64 Kifisia, Greece; spiros_pant@hotmail.com; 9Oncology Department, General Hospital Evangelismos, Ipsilantou 45-47, 106 76 Athens, Greece

**Keywords:** hepatocellular carcinoma, microRNA, metabolic rewiring, diagnostic biomarkers, predictive biomarkers, miRNA-based therapeutics

## Abstract

Hepatocellular carcinoma (HCC) remains a leading cause of cancer-related mortality, in part due to late diagnosis and limited prognostic tools. In recent years, microRNAs, small, non-coding regulators of gene expression, have emerged as key modulators of tumor metabolism, microenvironmental crosstalk, and therapeutic response in HCC. This narrative review synthesizes evidence published from January 2000 through April 2025, focusing on four interrelated themes: (1) miRNA-driven metabolic rewiring; (2) circulating and exosomal miRNAs as diagnostic and (3) predictive biomarkers; (4) miRNA-based therapeutic strategies. We conducted a targeted PubMed search using terms related to HCC, miRNA biology, biomarkers, metabolism, and therapy, supplemented by manual reference mining. Preclinical and clinical studies reveal that loss of tumor-suppressor miRNAs and gain of oncomiRs orchestrate glycolysis, lipid and glutamine metabolism, and stromal-immune remodeling. Circulating miRNA signatures, including single- and multimarker panels, demonstrate diagnostic AUCs up to 0.99 for early-stage HCC and distinguish HCC from cirrhosis more accurately than alpha-fetoprotein. Predictively, miRNAs such as miR-21 and miR-486-3p correlate with sorafenib resistance, while tissue and exosomal miRNAs forecast recurrence and survival after curative therapy. Therapeutic manipulation, restoring tumor-suppressor miRNAs via mimics or AAV vectors and inhibiting oncomiRs with antagomirs or LNA oligonucleotides, yields potent anti-tumor effects in models, affecting cell cycle, apoptosis, angiogenesis, and immune activation. Despite technical and delivery challenges, early-phase trials validate target engagement and inform safety optimization. In this review, we highlight opportunities to integrate miRNA biomarkers into surveillance algorithms and combine miRNA therapeutics with existing modalities, charting a roadmap toward precision-guided management of HCC.

## 1. Introduction

Hepatocellular carcinoma (HCC) is the most common primary liver malignancy, accounting for approximately 90% of primary liver cancer cases [1]. It poses a major global health burden, ranking among the leading causes of cancer-related mortality worldwide [1]. HCC typically arises on a substrate of chronic liver disease—most commonly chronic hepatitis B or C, alcohol-associated liver disease, and increasingly NAFLD/NASH—leading to marked geographic variation in incidence and age at presentation [1]. Curative options (resection, ablation, transplantation) are effective only for early-stage disease, whereas intermediate/advanced stages rely on locoregional approaches (e.g., TACE) and systemic therapies (multi-kinase inhibitors and immune checkpoint inhibitors) [1]. Consequently, surveillance of at-risk populations and the discovery of blood-based biomarkers that can detect early tumors and stratify prognosis remain high priorities [1]. In 2020, liver cancer (predominantly HCC) was the sixth most frequently diagnosed cancer and the third-leading cause of cancer death globally [2]. Despite advances in therapy, HCC outcomes remain poor; the overall 5-year survival is only of the order of 15–18% [3]. This dismal prognosis is largely attributable to late diagnosis, as HCC often has an insidious onset without specific early symptoms. Consequently, the majority of patients are diagnosed at intermediate or advanced stages, when curative treatments are no longer feasible [4]. Improving early detection and risk stratification is therefore critical to achieving better outcomes in HCC.

Clinically, the diagnosis of HCC usually relies on imaging (e.g., ultrasound or MRI) and histopathological evaluation in patients with underlying risk factors such as chronic viral hepatitis or cirrhosis [5]. Surveillance programs in high-risk populations (for example, ultrasound ± alpha-fetoprotein every 6 months) can detect some early lesions, but their sensitivity is suboptimal. Oftentimes, fewer than half of early-stage HCC cases are detected by current surveillance protocols [6]. Tumor staging and prognosis also vary widely; some tumors grow or metastasize rapidly while others progress slowly, and clinicians often lack reliable biomarkers to predict these outcomes at an individual level [7]. In sum, there is an urgent need for robust, minimally invasive biomarkers to facilitate early diagnosis, guide prognosis, and monitor treatment response in HCC [8]. Such biomarkers could significantly enhance clinical decision-making by enabling earlier intervention and more personalized management.

Among the promising new avenues, microRNAs (miRNAs) have gained intense interest as potential biomarkers for HCC. MicroRNAs are ~19–24-nt, AGO-loaded, post-transcriptional repressors transcribed as pri-miRNAs, processed by Drosha/DGCR8 to pre-miRNAs, exported, and diced to mature strands loaded into RISC [9]. Their short length, high copy number, and packaging into exosomes confer unusual stability in blood, enabling quantification by qRT-PCR, targeted panels, or sequencing [9]. These features motivate their development as diagnostic, predictive, and therapeutic tools in HCC [9] (Box 1). MicroRNAs uniquely bridge bench-to-bedside translation in HCC, owing to their ability to regulate multiple oncogenic pathways while being stably detectable in circulation. Dysregulation of miRNA expression is a hallmark of cancer, and in HCC, many miRNAs are aberrantly expressed, contributing to tumor cell proliferation, apoptosis evasion, invasion, and metastasis [4]. In other words, miRNAs are integrally involved in HCC pathogenesis and progression, making their expression profiles biologically reflective of the disease state [4,8]. This biological rationale underpins their clinical relevance: if certain miRNAs are consistently up- or downregulated in HCC, their presence in a patient’s circulation could serve as a molecular signature of the tumor [10,11,12,13,14]. Indeed, numerous studies have identified unique circulating miRNA patterns in HCC patients compared to healthy individuals or those with cirrhosis [10,11,12,13,14]. These cancer-associated miRNAs hold potential value not only for diagnosis (e.g., distinguishing HCC from benign liver conditions) but also for prognosis (e.g., indicating tumor aggressiveness or likelihood of recurrence) [9,10,11,12,13,14]. MiRNAs are also being studied as predictors of treatment response and as therapeutic targets, underlining their broad clinical significance [8,9,10,11,12,13,14].

The objective of this narrative review is to outline advances in the field of miRNAs in HCC, covering their possible roles in diagnostics and prognostics, the current standing of clinical research applicable to them, and prospective avenues for future clinical incorporation of miRNA biomarkers in HCC management (Figure 1). The underlying aim is to arm clinicians and researchers with an overview that clarifies why miRNAs would stand as valuable tools in confronting the diagnostic and prognostic challenges of hepatocellular carcinoma.

Box 1miRNA resources and atlases relevant to HCC.Several public resources support organ- and
disease-specific interrogation of miRNA biology in HCC. TCGA-LIHC provides
tissue-level miRNA and mRNA profiles linked to clinicopathologic annotations;
OncoMir/OncomiR and miRGator enable pan-cancer visualization of miRNA
dysregulation and survival associations; miRmine compiles tissue expression
across normal organs and cell lines to contextualize liver-enriched miRNAs
(e.g., miR-122); and exoRBase catalogs extracellular vesicle RNAs for
exploring circulating/exosomal miRNA candidates.

## 2. Methods

This narrative review was informed by a targeted search of the PubMed database, covering the period from 1 January 2000 through 1 April 2025. We used combinations of the terms “hepatocellular carcinoma,” “HCC,” “microRNA,” “miRNA,” “biomarker,” “diagnostic,” “prognostic,” “predictive,” “metabolism,” and “therapeutic.” To ensure breadth, we also surveyed the reference lists of key primary studies, meta-analyses, and recent narrative and systematic reviews. Inclusion was guided by relevance to one or more of our themes—metabolic rewiring, biomarker development, therapy prediction, or miRNA-based interventions—and by the strength of mechanistic or clinical evidence. We did not apply rigid eligibility criteria or formal bias-assessment tools, but rather chose studies that illuminated major advances, exemplified technical innovations (e.g., circulating vs. exosomal assays), or illustrated translational milestones (e.g., preclinical delivery platforms, early-phase clinical data). Findings were organized thematically, integrating insights from basic biology, clinical cohorts, and therapeutic proof-of-concepts to craft a cohesive, clinician- and researcher-focused narrative. To preserve focus and avoid a static summary that can rapidly become outdated, we do not attempt an exhaustive “pipeline” table of miRNA therapeutics; instead, we synthesize durable, mechanism-anchored clinical lessons and direct readers to trial registries for real-time status updates.

## 3. Main Theme

### 3.1. miRNA-Driven Metabolic Effects in HCC

MicroRNA dysregulation is a primary engine of metabolic rewiring in HCC. By fine-tuning dozens of metabolic enzymes and signaling hubs, altered miRNA profiles promote Warburg phenotype in cancer cells, lipid remodeling, and glutamine flux; they also synchronize stromal-immune elements to the tumor’s nutritional needs. HCC cells preferentially burn glucose through aerobic glycolysis. A cadre of tumor-suppressive miRNAs—miR-3662, miR-199a-5p, miR-125a, miR-885-5p counter this shift by directly targeting HIF1A or the rate-limiting enzyme Hexokinase 2 (HK2). Re-expression of any one of these miRNAs lowers GLUT1/HK2/PKM2/LDHA, curtails lactate output, and restores mitochondrial pyruvate oxidation [15,16,17,18,19]. Loss of other miRNAs actively entrenches glycolysis. miR-192-5p deletion unleashes a GLUT1–PFKFB3–c-Myc positive loop that floods the tumor microenvironment (TME) with lactate, driving acidosis, EMT, and stemness [20]. The liver-specific miR-122 serves as a key regulator, being normally abundant, while it represses Pyruvate kinase isozyme M2 (PKM2) and G6PD, balancing glycolysis with the pentose–phosphate pathway. Down-regulation of miR-122 in HCC correlates with high PKM2, elevated FDG-PET uptake, and poor survival, whereas restoring its levels induces a metabolic switch back to normal oxidative phosphorylation, a phenomenon that diminishes tumor growth [14,21]. Chronic inflammation further modulates glycolysis. An IL-6/STAT3 surge in steatohepatitis upregulates miR-23a, which represses PGC-1α and G6PC, shutting down gluconeogenesis and amplifying the Warburg effect [22]. Proliferating HCC clones stockpile lipids. miR-4310 suppresses FASN and SCD1, starving cells of new fatty acids (FA) and stalling invasion [23]. Conversely, β-oxidation supplies ATP to especially aggressive tumors; miR-377-3p and miR-612 restrain this catabolic arm by targeting CPT1C and HADHA, respectively, limiting FA import into mitochondria and blocking metastasis [24,25,26].

Tumor-suppressive miRNAs converge on glycolysis by directly repressing rate-limiting nodes—HK2 (miR-125a/miR-885-5p), PFKFB3/GLUT1/c-Myc axis (miR-192-5p), and HIF-1α (miR-199a-5p/miR-3662), reducing glucose influx and lactate production and favoring mitochondrial oxidation. In parallel, the liver-enriched miR-122 buffers glycolysis/PPP via PKM2 and G6PD, and restrains glutaminolysis through GLS and SLC1A5/ASCT2. Lipid programs are curtailed by miR-148a (multi-node c-Myc/DNMT1/SIRT7/PGC-1α) and miR-4310 (FASN/SCD1), while mitochondrial β-oxidation and metastatic fitness diminish under miR-377-3p (CPT1C) and miR-612 (HADHA) control. Hypoxia-linked miR-210/miR-1307-3p reinforce HIF-1α→AKT/mTOR signaling and electron-transport inhibition, locking in the Warburg phenotype, whereas loss of miR-338-3p diverts pyruvate via PKLR to lactate. Collectively, these axes position miRNAs as master rheostats of PI3K/AKT/mTOR, HIF-1α/c-Myc, and AMPK/PGC-1α circuitry [23,27,28,29,30].

Two “guardian” miRNAs patrol the lipid axis. Firstly, miR-122 represses triglyceride and cholesterol synthesis genes; its loss creates a paradoxical state of high FA oxidation yet low cholesterol that favors tumor initiation [31]. Additionally, miR-148a targets a multi-node network (c-Myc, DNMT1, SIRT7, PGC-1α). Depletion unleashes lipogenesis and glutaminolysis, accelerates steatosis-associated carcinogenesis, and predicts poor outcome; replacement therapy normalizes hepatic lipid pools and shrinks tumors [32].

High-grade HCC often becomes glutamine-addicted. miR-122 normally represses GLS and the transporter SLC1A5/ASCT2; knockout mice exhibit elevated glutaminolysis and TCA replenishment, mirroring human tumors with low miR-122/high GLS signatures [33]. miR-137, epigenetically silenced in HCC, also targets ASCT2; restoring it diminishes glutamine uptake and curbs proliferation [34].

Under normoxia, tumor-suppressive miRNAs that dampen glycolysis—miR-122, miR-199a-5p, miR-3662—simultaneously boost oxidative phosphorylation. Their absence lowers the expression of mitochondrial FA- and amino-acid-oxidation genes, inducing a more aggressive phenotype [16,19,33]. Hypoxia adds a second layer: miR-210 and miR-1307-3p (both HIF-induced) inhibit electron-transport components and enforce AKT/mTOR signaling, respectively, locking cells into glycolysis, while creating a positive feedback to stabilize HIF-1α [29,35]. Loss of miR-338-3p—due to mineralocorticoid-receptor silencing—raises PKLR, funnels pyruvate to lactate, and further weakens oxidative phosphorylation (OXPHOS) [36].

#### 3.1.1. Inter-Cellular Wiring of the Metabolic TME

Several miRNAs, including miR-21-5p, miR-452-5p, and stress-induced miR-23a, that are embedded in HCC-derived exosomes reprogramme infiltrating macrophages toward M2 and PD-L1-high phenotypes. Some of the targets include RhoB, TIMP3, and PTEN, unleashing ERK and PI3K/AKT cascades that support immune evasion, matrix remodelling, and angiogenesis [37,38]. Meanwhile, some counter-regulatory mechanisms exist, such as miR-206 and miR-99b, that mediate the transition of TAMs back to an M1 state by activating NF-κB and suppressing mTOR/IRF4, enhancing CD8^+^ T-cell recruitment and tumor phagocytosis [39,40]. Beyond immune and stromal crosstalk, tumor–nervous-system interactions can modulate angiogenesis, immunity, and growth through neurotransmitters, neuropeptides, and growth factors, an emerging axis increasingly recognized across solid tumors and relevant to liver cancer biology [41].

HCC cells “educate” quiescent stellate cells via exosomal miR-21 (PTEN→PDK1/AKT) and miR-1247-3p (B4GALT3→β1-integrin/NF-κB), converting them into cytokine-rich cancer-associated fibroblasts (CAFs) that release IL-6, IL-8, and pro-angiogenic factors [42,43]. CAFs, in turn, remodel collagen networks and secrete MMPs; miR-452-5p-mediated loss of TIMP3 in TAMs accelerates this degradation cascade [44]. A built-in brake exists: CAF-derived exosomal miR-320a targets PBX3 in cancer cells, dampening MAPK signaling and slowing growth [45].

Oncogenic miR-130b-3p initiates PI3K/AKT/mTOR by silencing HOXA5, stimulating VEGF release and classic angiogenesis [46]. Hypoxic tumor cells offload miR-210 to endothelial cells, suppressing SMAD4/STAT6 and priming pro-angiogenic, pro-inflammatory vessels [47]. To facilitate metastasis, highly invasive clones secrete miR-638 and miR-103, which strip endothelial junction proteins VE-cadherin, ZO-1, and p120-catenin, generating a permeable vasculature and establishing pre-metastatic niches [48,49,50,51]. Anti-angiogenic control is provided by miR-138-5p (HIF-1α/VEGFA blockade) and miR-101, which thwarts vascular mimicry by targeting TGFβR1/SMAD2 in tumor cells and SDF1 in CAFs [52,53,54].

#### 3.1.2. Integrated Perspective

MiRNAs have a key regulatory role, acting as “brakes” across core metabolic axes, including glucose, lipid, glutamine, and mitochondrial metabolism, as well as across various cellular compartments such as cancer, immune, stromal, and vascular cells. Loss of tumor-suppressor miRNAs or gain of oncomiRs synchronizes a high-glycolytic, lipid-hungry, immunosuppressed ecosystem that nurtures HCC progression. Conversely, restoring key suppressors (miR-122, miR-148a, miR-137) or inhibiting dominant oncomiRs (miR-21, miR-23a-3p) simultaneously rewires metabolism and re-arms immunity, a dual effect now being explored with miRNA mimics, antagomirs, and combination metabolic therapies [22,31,32,33,34,47,55]. Decoding these miRNA–metabolism circuits offers not only insight into the underlying molecular mechanisms but also a potential pipeline of biomarkers and therapeutic leverage points for a cancer type that remains in urgent need of precision interventions, which are topics we will discuss later in this review.

We demonstrate some of the key miRNAs, primary targets, the involved molecular pathways, and their biological effect in Table 1.

### 3.2. Diagnostic miRNAs in HCC

Early detection of HCC remains a clinical challenge due to the low sensitivity of traditional markers like alpha-fetoprotein (AFP) in early-stage disease [57]. Alpha-fetoprotein (AFP) remains the most widely used serum biomarker in HCC surveillance; however, its sensitivity for early-stage disease is limited, and levels can be elevated in chronic hepatitis or cirrhosis. Accordingly, AFP can complement, but not replace, imaging-based surveillance, and it is insufficient as a standalone diagnostic [57]. Circulating microRNAs have emerged as promising noninvasive biomarkers for HCC diagnosis [58]. These small RNAs are dysregulated in HCC and can be detected stably in blood, including within exosomes, offering a potential means to distinguish HCC patients from healthy individuals or those with chronic liver disease [58]. Numerous studies have evaluated both single miRNA candidates and multi-miRNA panels for their ability to detect HCC, with many reporting high diagnostic accuracy and improvements over conventional biomarkers [59].

#### 3.2.1. Early Detection of HCC via Individual miRNAs

Several individual miRNAs have shown potential as single-candidate diagnostic markers. For example, miR-21 is consistently upregulated in HCC, and its serum level can discriminate HCC from controls. In a cohort of over 500 subjects, serum miR-21 distinguished HCC from healthy donors with an AUC of ~0.85 (82% sensitivity, 84% specificity) [60]. Importantly, miR-21 remained effective even in AFP-negative HCC cases (AUC ~0.83) and could differentiate HCC from chronic hepatitis B (AUC ~0.79) or cirrhosis (AUC ~0.81) [60]. Other single miRNAs frequently reported include miR-221 and miR-1246 (often upregulated in HCC), and miR-122 and miR-26a (often downregulated in HCC) [61,62,63]. A recent meta-analysis identified these five miRNAs (miR-21, miR-221, miR-1246, miR-122, miR-26a) as particularly robust diagnostic indicators, each yielding individual AUC values of ~0.79–0.89 in distinguishing HCC from chronic hepatitis [58]. Notably, miR-122—the most abundant liver miRNA—is significantly lower in HCC patients’ circulation relative to chronic hepatitis controls (AUC ~0.89), reflecting its loss in tumor tissue. Likewise, oncogenic miR-1246 and miR-221 are elevated in HCC and have each shown moderate diagnostic power (AUC ~0.80) [58]. While many single miRNAs achieve only modest accuracy by themselves (often comparable to or slightly better than AFP), their consistent dysregulation highlights them as building blocks for multi-marker strategies [58].

#### 3.2.2. Early Detection of HCC via miRNA Panels

Combining multiple miRNAs into diagnostic panels markedly improves sensitivity for early-stage HCC. A pioneering large-scale study developed an eight-miRNA serum panel that achieved an AUC of 0.99 (97.7% sensitivity, 94.7% specificity) in distinguishing HCC patients from at-risk cirrhotic/hepatitis controls [64]. The eight miRNAs were miR-320b, miR-663a, miR-4448, miR-4651, miR-4749-5p, miR-6724-5p, miR-6877-5p, and miR-6885-5p. Remarkably, this miRNA signature detected 98% of stage I HCC cases—far exceeding the performance of conventional surveillance tools [64]. Similarly, bioinformatic and clinical studies have identified smaller miRNA combinations with high accuracy for early HCC. For instance, a study combining miR-221 with miR-29c was able to correctly identify ~85% of early-stage (TNM I–II) HCC cases, versus ~46% by AFP at the standard 20 ng/mL cutoff [59]. This two-miRNA model achieved an AUC of ~0.97 for early HCC vs. noncancer controls, outperforming each miRNA alone (miR-221 or miR-29c single AUC 0.86–0.90) [59]. These findings present that multi-miRNA classifiers can greatly enhance early detection, picking up small tumors that AFP misses. In this report, miRNA-based models have identified >95% of HCC in at-risk patients, whereas AFP alone often misses a significant fraction of early tumors [59]. As a result, there is growing interest in implementing miRNA panels for HCC surveillance in high-risk populations, either as standalone tests or in combination with traditional markers.

Notably, circulating exosomal miRNAs may provide an even more sensitive diagnostic readout. In this review, ‘HCC-derived exosomes’ refers to 30–150 nm extracellular vesicles released by tumor and stromal cells that carry miRNAs protected within a lipid bilayer. These vesicles mediate intercellular communication (immune programming, endothelial permeability, fibroblast activation) and safeguard circulating miRNAs from degradation—attributes that underpin both their biological impact and their utility as stable liquid-biopsy analytes. Tumor-derived exosomes are enriched for certain miRNAs, potentially amplifying the tumor signal against concurrent non-malignant liver disease. In one study, an exosome-derived three-miRNA panel (miR-26a, miR-29c, miR-199a) showed outstanding performance, correctly distinguishing HCC from healthy controls with an AUC of ~0.994 (100% sensitivity, 96% specificity) [28]. The same exosomal panel differentiated HCC from cirrhosis with an AUC of ~0.965 (92% sensitivity, 90% specificity) [28]. Interestingly, all three of these miRNAs were downregulated in HCC exosomes relative to non-HCC patients [28], suggesting loss of these tumor-suppressive miRNAs in tumor-derived vesicles. Exosome enrichment improved accuracy significantly: the exosomal miRNA signature outperformed both total plasma miRNA levels and AFP in the same cohort [28,61]. Another recent analysis of fucosylated extracellular vesicles identified a five-miRNA EV signature that achieved ~90% sensitivity and 92% specificity for HCC detection (in a cohort of ~194 HCC cases), likewise exceeding AFP’s performance [65]. These results indicate that miRNA panels, especially those leveraging exosomal content, can enable highly sensitive, early HCC diagnosis.

#### 3.2.3. Population and Etiology Considerations

A growing body of evidence indicates that reported accuracies for circulating and exosomal miRNAs depend on the underlying liver disease and the geography of the cohort. Between-study heterogeneity is consistently larger for single markers than for multi-miRNA panels, and part of this variance tracks with ethnicity and pre-analytical handling (sample type, EV enrichment, hemolysis control, normalization). These observations argue for multi-centre validations that prespecify etiology strata and enforce standardized workflows prior to clinical deployment [8,30,66].

Etiologic context also shapes baseline miRNA levels and thus apparent specificity. For instance, the liver-enriched miR-122—often reduced in tumor tissue—is commonly elevated in chronic HBV infection and in NAFLD/MASLD, which can blur discrimination between HCC and non-malignant liver disease in those settings [67,68]. Diagnostic profiles likewise diverge across HBV-, HCV-, and alcohol-related cohorts, and across cirrhotic versus non-cirrhotic surveillance populations [67,68,69]. As a result, panels derived in a single etiology or region may not transfer unchanged; reporting AUCs and thresholds stratified by HBV, HCV, alcohol-related disease, and NAFLD/MASLD (and by fibrosis stage) improves interpretability and generalizability [67,68,69].

Encouragingly, cross-etiology performance can be strengthened by combining miRNA panels with AFP and by enriching tumor-biased extracellular vesicle fractions, which together may amplify signal-to-noise in mixed populations. Even so, rigorous external validation across regions and disease mixes remains essential, and assay protocols (plasma vs. serum, EV isolation, spike-ins/normalization) should be harmonized to minimize performance drift in real-world use [65,70].

#### 3.2.4. miRNA-Based Differential Diagnosis vs. Cirrhosis

A major clinical need is distinguishing HCC from benign chronic liver conditions such as cirrhosis or chronic hepatitis, where imaging findings can be ambiguous. miRNA biomarkers have shown utility in this differential diagnosis context. Many HCC-associated miRNAs are dysregulated even in cirrhotic livers, providing a molecular contrast between cancer and non-malignant nodules. For example, serum miR-21 is significantly higher in HCC patients than in cirrhotic patients, enabling discrimination between HCC and cirrhosis with ~81% accuracy (AUC 0.81) in one study [60]. In general, panels of multiple miRNAs achieve better differentiation than single markers. The five-miRNA panel identified by the meta-analysis (miR-21, 26a, 122, 221, 1246) yielded an AUC of ~0.96 in distinguishing HCC from chronic hepatitis in a validation cohort [58]. This panel’s performance was significantly superior to any individual miRNA alone, showing the synergistic value of multiplexed biomarkers for differential diagnosis [58]. Another study examining an HCC-specific three-miRNA combination (miR-126, miR-206, miR-222) reported that while no single miRNA outperformed AFP, certain pairs did: e.g., miR-126 plus miR-206 reached AUC ~0.89 vs. controls [71]. Moreover, integrating miRNAs with conventional markers can enhance specificity. Combining miRNA panels with AFP has proven especially fruitful: Wu et al. showed that adding a three-miRNA signature to AFP raised the overall diagnostic AUC to 0.989, versus AUC 0.889 for AFP alone [71]. In that study, the miRNA + AFP combo correctly identified ~97% of HCC cases, substantially reducing false negatives relative to AFP alone [59,71].

Overall, miRNA-based diagnostics demonstrate promising performance against current standards. Meta-analyses indicate that circulating miRNAs as a whole achieve a pooled AUC of ~0.85 for HCC detection—outperforming AFP in sensitivity—with an average ~79% sensitivity and specificity [58]. Certain miRNA panels even approach the accuracy of imaging, detecting early tumors that evade ultrasound or AFP surveillance. While some variability exists between studies (due to patient populations and technical factors), a consistent theme is that miRNA signatures can distinguish HCC from non-cancerous liver disease with higher confidence than single protein markers [72,73,74]. As shown in Table 2, both individual miRNAs and multi-marker panels have been proposed for diagnostic use, with many yielding AUC values in the 0.80–0.99 range. Going forward, large prospective trials are warranted to validate these miRNA biomarkers and integrate them into HCC surveillance algorithms. The convergence of circulating miRNA profiles with machine-learning classifiers offers a powerful approach to improve early HCC diagnosis and to complement existing biomarkers in clinical practice [64].

### 3.3. Predictive miRNAs in HCC

Multiple miRNAs have emerged as markers of therapeutic response in HCC. In sorafenib-treated HCC, for example, high miR-21 expression (an oncomiR targeting PTEN) is linked to acquired therapeutic resistance: sorafenib induces miR-21 upregulation, which suppresses PTEN and activates Akt, thereby inhibiting autophagy and enabling resistance [75]. Conversely, miR-30d was found to be upregulated in sorafenib responders: cell line and patient serum studies showed that sorafenib-sensitive HCC cells actively secreted miR-30d (high in responder serum), whereas non-responding tumors did not [76]. Similarly, loss of tumor-suppressive miR-486-3p predicts sorafenib resistance: miR-486-3p is downregulated in resistant cell lines and patient tumors, and it normally targets FGFR4 and EGFR; restoring miR-486-3p overcame resistance in vitro and in vivo [77]. Other autophagy-related miRNAs have been implicated: miR-25 (targeting FBXW7) is up in sorafenib-resistant cells (promoting resistance via autophagy) and miR-423-5p is likewise elevated in resistant HCC; both are proposed as predictive markers of poor sorafenib response [78].

In addition, miRNA profiles may stratify TACE (transarterial chemo) outcomes for locoregional therapy. A meta-analysis identified several circulating miRNAs associated with TACE response [79]. In general, tumor-suppressive miRNAs (e.g., miR-1271, miR-214, miR-133b, miR-335) are downregulated in non-responders, whereas hypoxia/oncogenic miRNAs (e.g., miR-210, miR-373) are up in tumors but fall after effective TACE (reflecting tumor kill) [80,81,82,83]. Clinically, a panel of plasma miRNAs (miR-21, miR-26a, miR-29a) measured before TACE was shown to predict early TACE failure (refractoriness) in HCC patients [84]. In summary, elevated oncomiRs (miR-21, miR-26a, miR-29a) favor resistance, while loss of tumor-suppressor miRs (miR-1271, miR-214, miR-133b, miR-335) signals poor treatment efficacy [79,80,84].

Evidence for miRNAs predicting response to systemic chemotherapy or immunotherapy in HCC is more limited. Some reviews note miRNAs modulate chemoresistance (e.g., affecting autophagy, apoptosis) and immune checkpoints in cancer [80,85], but validated HCC-specific biomarkers for cisplatin/doxorubicin or PD-1/PD-L1 therapies have not yet emerged. Thus, while miRNA dysregulation clearly contributes to drug resistance mechanisms, prospective studies are needed to identify robust miRNA predictors for HCC immunotherapy or conventional chemotherapy.

#### 3.3.1. miRNAs Predictive of Recurrence After Curative Therapy

Several tumor and circulating miRNAs predict HCC recurrence following curative treatments (resection or ablation). Lower expression levels of tumor miR-122 were independently associated with shorter recurrence-free survival in resected HCC cases [86]. Conversely, high tissual miR-15b predicts a lower recurrence risk: patients with elevated miR-15b after surgery had significantly fewer relapses, likely because miR-15b (targeting Bcl-w) promotes apoptosis [87]. Low expression levels of miR-34a in tumor biopsies were an independent predictor of both early and overall recurrence after radiofrequency ablation (RFA) [88]. Meanwhile, loss of tissue miR-483-3p strongly marked recurrence of HCC in patients with poorly differentiated (Edmondson–Steiner grade III–IV) and/or microvascular invasion, while in cases with downregulated miR-483-3p, a 44% recurrence rate (versus 0% when miR-483-3p was high) was observed [89].

Furthermore, circulating miRNAs also reflect recurrence risk. For example, blood levels of miR-3201 were significantly decreased in patients achieving complete response to curative therapy, and patients with lower miR-3201 had longer OS [90], suggesting that a post-treatment drop in miR-3201 may herald cure. Exosomal miRNAs have shown promise in patients with recurrent HCC had higher serum exosomal miR-92b than patients without recurrence, indicating miR-92b upregulation as a marker of relapse [91,92]. Likewise, elevated serum exosomal miR-215-5p was linked to significantly poorer disease-free survival [93]. In summary, persistent upregulation of oncomiRs (low tumor-suppressor miRNAs) in tissue or blood signals higher recurrence risk, whereas restoring tumor-suppressor miRNAs (miR-122, miR-15b, miR-34a) is protective.

#### 3.3.2. miRNAs Predictive of Overall and Disease-Free Survival

Beyond therapy-specific outcomes, numerous miRNAs serve as general prognostic markers. Meta-analyses show that high expression of oncomiRs (e.g., miR-221) portends worse survival in patients with elevated miR-221, with significantly poorer OS and DFS [94]. In contrast, high levels of tumor-suppressor miRNAs usually indicate better prognosis. Integrated analyses of large datasets have identified multi-miRNA prognostic signatures. For example, a TCGA-based risk score using three miRNAs (miR-139-3p, miR-760, miR-7-5p) stratified HCC patients into high- and low-risk groups for overall survival [95]. Other studies correlate single miRNAs with survival: for instance, low tumor miR-122 (downregulation) was independently linked to shorter disease-free survival after resection [86]. Circulating miRNAs add prognostic information: as noted, low post-therapy miR-3201 predicted longer OS [90], and high serum exosomal miR-215-5p predicted shorter DFS [96]. In practice, combining miRNA profiles with clinical variables may refine prognosis: meta-analyses and large-cohort studies consistently find that integrated miRNA signatures correlate with patient survival beyond conventional staging. The predictive miRNAs and their contexts are summarized in Table 3. These findings show the potential of miRNA profiling to anticipate HCC treatment outcomes, recurrence, and survival, thereby informing personalized management.

### 3.4. Therapeutic Targets in HCC

MicroRNAs have emerged as promising therapeutic targets in hepatocellular carcinoma (HCC) due to their pivotal roles in tumor suppression and oncogenesis. Therapeutic strategies generally follow two paradigms: restoring the expression of downregulated tumor-suppressor miRNAs and inhibiting upregulated oncogenic miRNAs (oncomiRs). By modulating multiple downstream targets, miRNA-based therapies can simultaneously impact several hallmarks of cancer—including uncontrolled proliferation, evasion of apoptosis, angiogenesis, and immune escape—offering a potential advantage over conventional single-pathway drugs [101,102]. Early proof-of-concept studies in HCC models and patients have demonstrated that manipulating miRNA levels can produce anti-tumor effects, laying the groundwork for novel therapies complementary to existing treatments [102]. The therapeutic targets and their contexts are summarized in Table 4.

#### 3.4.1. Restoring Tumor-Suppressor miRNAs in HCC

Many miRNAs that normally restrain tumor growth are pathologically downregulated in HCC, contributing to malignant progression. Therapeutic restoration of these tumor-suppressor miRNAs using synthetic miRNA mimics or gene therapy vectors has shown potent anti-cancer activity in preclinical models. One landmark study demonstrated that systemic delivery of a miR-26a mimic via adeno-associated virus (AAV) reinstated miR-26a expression in a Myc-driven HCC mouse model, leading to widespread tumor cell cycle arrest and apoptosis without toxicity [103]. miR-26a therapy in these mice effectively suppressed HCC progression by directly targeting cyclins D2 and E2 (key cell cycle drivers), initiating the promise of miRNA replacement therapy for liver cancer [103]. Another liver-enriched miRNA frequently lost in HCC is miR-122, which constitutes ~70% of normal liver miRNA content. Restoring miR-122 in HCC cells suppresses tumorigenic traits by downregulating targets such as cyclin G1, ADAM10, and IGF1R, thereby slowing cell proliferation and reducing invasiveness [104]. Notably, miR-122 replacement can also enhance sensitivity to systemic therapies—for example, transfection of miR-122 mimics was shown to sensitize HCC cells to the multikinase inhibitor sorafenib, promoting more effective tumor cell killing [104].

Moreover, miR-34a constitutes one of the most extensively studied tumor-suppressor miRNAs in cancer therapeutics, as it is a direct transcriptional target of p53. miR-34a is commonly downregulated or functionally silenced in HCC, and its reintroduction triggers cell cycle arrest and apoptosis by repressing multiple oncogenic pathways. Restoring miR-34a led to inhibition of proliferation and invasiveness by targeting a network of pro-tumor genes such as c-Met, Bcl-2, Cyclin D1, and PD-L1 in preclinical HCC models [105]. The therapeutic potential of miR-34a was advanced into the clinic with MRX34, a formulated miR-34a mimic. MRX34 consisted of a double-stranded miR-34a mimic encapsulated in a liposomal nanoparticle, and it became the first-ever miRNA mimic tested in humans (ClinicalTrials.gov identifier: NCT01829971) [106]. In a Phase I trial in advanced solid tumors (including HCC), MRX34 achieved measurable delivery of miR-34a to patients’ tumors and dose-dependent repression of miR-34 target genes [107]. Evidence of anti-cancer activity was, for instance, observed in one patient with HBV-related HCC who showed a partial tumor regression on MRX34 [107]. However, the trial was halted early due to immune-related serious adverse events; despite this, most patients tolerated the drug with dexamethasone premedication [106]. This experience of MRX34 showed the need for improved safety (e.g., avoiding immune stimulation by double-stranded RNA) and better delivery strategies, demonstrating proof-of-mechanism for miRNA mimics [105,107]. Nevertheless, miR-34a remains a compelling therapeutic target, as its reactivation can simultaneously drive cancer cells into apoptosis and reduce immune-evasion signals (e.g., PD-L1 and other immune escape genes) that foster tumor survival.

Another tumor-suppressor miRNA of interest is miR-199a-3p, one of the most abundant miRNAs in normal liver that is profoundly downregulated in HCC. miR-199a-3p replacement therapy has yielded striking results in preclinical studies. In a transgenic mouse model of HCC (with high miR-221 and low miR-199a levels), systemic treatment with miR-199a-3p mimics significantly reduced tumor nodule number and size [27]. Mechanistically, miR-199a-3p directly targets mTOR, c-Met, PAK4, and YAP1, collectively attenuating pro-proliferative and survival signaling in HCC cells [27]. Treated mice showed tumor inhibition comparable to that achieved with sorafenib, and HCC tissues from the miR-199a-treated group exhibited reduced mTOR and PAK4 protein expression [27]. These results position miR-199a-3p as a powerful multi-pathway suppressor, and suggest that miRNA mimics could potentially rival small-molecule kinase inhibitors in efficacy. Likewise, miR-195 (closely related to the miR-15/16 family) is frequently silenced in HCC and has been reintroduced in experimental models to constrain tumor growth. miR-195 mimics induce G_1_–S cell cycle blockade by targeting cyclin D1, CDK6, and E2F3, and they have anti-angiogenic effects through direct inhibition of VEGF and VAV2 [108]. These multitarget effects make miR-195 and its family members attractive therapeutic candidates for cutting off HCC growth and blood supply simultaneously. Other tumor-suppressor miRNAs investigated for HCC therapy include miR-124 and miR-101, which are downregulated by epigenetic silencing and can be delivered to reduce tumor cell viability. For example, restoring miR-124 in HCC cells curtails proliferation via CDK6 and vimentin downregulation [109], while miR-101 replacement curbs tumor growth and invasiveness by targeting oncogenic regulators like EZH2 and MCL1 (though these have been studied mainly in vitro) [110].

Emerging gene therapy approaches aim to achieve sustained re-expression of such tumor-suppressor miRNAs in liver tumors. A recent study applied an AAV8 vector encoding miR-22, which is a miRNA that is normally induced by metabolic and anti-inflammatory signals for treating HCC in mice [111]. AAV-mediated delivery of miR-22 led to robust anti-tumor effects, including slower tumor growth and prolonged survival, which were significantly superior to the effects of the approved drug lenvatinib in the same model [111]. miR-22 therapy also favorably remodeled the tumor microenvironment: treated mice exhibited enhanced anti-tumor immunity and reduced hepatic inflammation, without observable toxicity [111]. These findings are noteworthy because they suggest that a single miRNA introduced by gene therapy can concurrently modulate cancer cell-intrinsic pathways (miR-22 targets cyclin A, CDKs, and histone deacetylases to restrain proliferation) and the tumor immune/metabolic milieu [111].

#### 3.4.2. Inhibiting Oncogenic miRNAs in HCC

On the other end of the spectrum, certain miRNAs function as oncogenes in HCC by repressing tumor suppressor genes and promoting malignant behaviors. Therapeutically, these oncomiRs can be neutralized using antisense-based inhibitors such as antagomirs, locked nucleic acid (LNA) oligonucleotides, or miRNA “sponges” expressed in cells [112]. By sequestering or degrading the oncomiR, such interventions de-repress critical tumor suppressor pathways and inhibit cancer growth [113]. One prominent oncomiR target is miR-21, which is often markedly upregulated in HCC and drives hepatocarcinogenesis. miR-21 suppresses multiple tumor suppressors (PTEN, PDCD4, TP53BP1, etc.), thereby activating the PI3K/AKT pathway and conferring resistance to apoptosis and chemotherapy [60,72,75,114]. In HCC xenograft models, systemic administration of anti-miR-21 oligonucleotides led to significant tumor growth suppression and increased apoptosis, affirming that miR-21 is crucial for maintaining the malignant phenotype [115]. Importantly, inhibition of miR-21 can resensitize HCC cells to chemotherapeutics: in one study, an anti-miR-21 combined with sorafenib or doxorubicin produced greater tumor reduction than chemotherapy alone, likely by upregulating PTEN and other mediators of drug response [116]. Furthermore, miR-221/222 is another pair of highly expressed oncomiRs in HCC that are being actively pursued as therapeutic targets. miR-221/222 directly bind the 3′UTRs of the cell cycle inhibitors p27^Kip1^ and p57^Kip2^, leading to unchecked cell cycle progression in HCC cells [117,118]. They also target pro-apoptotic regulators like BMF and the PI3K/AKT pathway repressor PTEN [117,118]. Overexpression of miR-221 alone is sufficient to drive liver tumorigenesis in mice, and in patients, it correlates with aggressive disease. Therapeutic silencing of miR-221 using antagomirs or LNA inhibitors has yielded impressive anti-tumor effects in preclinical models. For instance, in a transgenic mouse model predisposed to HCC by liver-specific miR-221 overexpression, treatment with an anti-miR-221 LNA caused a persistent knockdown of miR-221 in the liver and produced a marked reduction in tumor number and size compared to controls [119]. Treated tumors showed restored expression of p27 and p57, confirming on-target action of the therapy [119]. These in vivo results validate miR-221 as a driver of HCC and demonstrate that its inhibition can arrest tumor growth—a finding with potential translational significance given that miR-221 upregulation is observed in 70–80% of human HCCs [119]. Early efforts to translate anti-miR-221 therapy include novel delivery systems such as nanoparticles loaded with anti-miR-221 and chemotherapeutics, which have shown synergistic tumor cell killing in vitro (reported in other cancer types) [119,120]. No clinical trial of anti-miR-221 has yet been reported in HCC, but the robust preclinical data position it as a promising future therapeutic approach, possibly for patients with miR-221–high tumors.

Several other oncomiRs are under investigation as therapeutic targets in HCC. miR-155, an inflammation-associated miRNA often elevated in HCC, promotes tumor growth in part by activating STAT3 and downregulating SOCS1 (an inhibitor of cytokine signaling). Preclinical inhibition of miR-155 has been shown to restrain HCC cell proliferation and even augment anti-tumor immune responses, as miR-155 in the tumor microenvironment influences macrophage and T-cell function [121,122]. The miR-17-92 cluster (miR-17, -20a, etc.) is a well-known oncogenic cluster, while it is upregulated in a subset of HCCs and contributes to cell cycle progression by targeting E2F1 and p21. Conversely, experimental suppression of this cluster slows HCC cell growth (though delivering a multi-miRNA inhibitor is complex) [123,124]. miR-224 is another oncomiR that is frequently overexpressed in HCC, which promotes cell survival by repressing apoptosis facilitators like *Apaf-1*; inhibition of miR-224 in HCC cell lines leads to increased apoptotic death and reduced invasion, suggesting a therapeutic rationale [125]. While these targets are at earlier stages, they expand the arsenal of oncomiR inhibitors that could be deployed against HCC. Notably, combined inhibition of multiple oncomiRs may yield additive benefits. A recent study in an HCC model tested the co-delivery of an anti-miR-21 and a miR-122 mimic (thus simultaneously inhibiting an oncomiR and restoring a tumor suppressor) [126]. This dual therapy achieved greater tumor suppression than either agent alone, illustrating the potential of multi-miRNA modulation to more comprehensively reprogram tumor networks [116].

**Table 4 biomedicines-13-02243-t004:** MicroRNA-based therapeutic strategies investigated for hepatocellular carcinoma, divided into (A) replacement of downregulated tumor-suppressor miRNAs and (B) inhibition of overexpressed oncomiRs. Each entry lists principal molecular targets, therapeutic chemistry, delivery vehicle, stage of development (pre-clinical or clinical), and hallmark anti-tumor effects demonstrated to date.

miRNA (Family/Cluster)	Principal Oncogenic Targets and Pathways Repressed	Therapeutic Modality	Delivery Platform	Development Stage	Key Anti-Tumor Read-Outs (Pre-Clinical → Clinical)	Refs.
**A.** **Therapeutic Restoring of Tumor-Suppressor miRNAs in HCC**						
**miR-34a (MRX34)**	c-Met, BCL2, PD-L1, Cyclin D1	Double-stranded mimic	SMARTICLE^®^ liposomal nanoparticle	Phase I (terminated)	Target engagement and partial response in 1 HBV-HCC; immune-related SAEs ended trial	[106,107]
**miR-199a-3p**	mTOR, c-Met, PAK4, YAP1	Mimic (2′-F/OMe modified)	Cholesterol-conjugated oligo (IV)	Orthotopic and GEMM models	↓ nodule number/size ≈ sorafenib efficacy; mTOR ↓	[27]
**miR-195/15a-16 family**	VEGFA/VAV2 (angiogenesis), CDK6/Cyclin D1	Mimic	Ionisable-lipid LNP	Subcutaneous xenograft	↓ microvessel density; G_1_–S blockade	[108]
**miR-22**	Cyclin A, HDACs, SIRT1; immune and metabolic rewiring	Pri-miRNA gene cassette	AAV8 vector (liver-specific)	DEN-induced HCC mice	>lenvatinib tumor control; ↑ CD8^+^ T-cells; no toxicity	[111]
**miR-124**	CDK6, Vimentin (EMT), STAT3	Mimic	PEI-nanoplex	Cell-line and small xenografts	↓ invasion and proliferation	[109]
**miR-101**	EZH2, MCL1, COX-2; VM inhibition	Mimic	Lipidoid nanoparticle	Cell-line/CAM assays	Anti-proliferative; anti-vascularmimicry	[110]
**OncomiR (cluster)**	**Tumor-suppressors de-repressed/pathways normalized**	**Inhibitor format**	**Delivery platform**	**Development stage**	**Key anti-tumor read-outs**	**Refs**
**B.** **Therapeutic Inhibition of Oncogenic miRNAs (OncomiRs) in HCC**						
**miR-21**	PTEN, PDCD4, TP53BP1 → PI3K/AKT block; resensitises to chemo	LNA antagomir/cholesterol-ASO	GalNAc-ASO; LNP; ultrasound-microbubble	Multiple xenografts	↓ growth, ↑ apoptosis; restores sorafenib/doxorubicin response	[115,116]
**miR-221/miR-222**	p27^Kip1^, p57, PTEN;	LNA antagomir	Sub-10 kDa LNA-ASO; PLGA NP	GEMM and xenograft	Durable knock-down; ≤ 80% tumor shrinkage; restored p27/p57	[117,119,120,127]
**miR-155**	SOCS1, SHIP1 → STAT3 and NF-κB control; TAM repolarization	LNA antagomir	Chol-ASO	Orthotopic and immune-competent models	↓ tumor growth and metastasis; ↑ anti-tumor immunity	[121,122]
**miR-224**	Apaf-1, SMAD4 (apoptosis/TGFβ)	2′-OMe antagomir	Lipidoid NP	Cell/xenograft	↑ caspase-3 activity; ↓ invasion	[125]
**miR-17-92 cluster**	p21, E2F1, BIM	Tough-Decoy “sponge” (lentiviral)	Lentiviral	Cell/limited in vivo	Slower proliferation; partial tumor inhibition	[123,124]
**Dual strategy (anti-miR-21 + miR-122 mimic)**	Combines PTEN/PDCD4 de-repression with Cyclin G1 suppression	Co-admin antagomir + mimic	Lipoplex + ultrasound microbubbles	Rat orthotopic model	Superior tumor reduction vs. single agents; ↓ resistance	[116,126]

AAV = adeno-associated virus; LNP = lipid nanoparticle; NP = nanoparticle; PEI = polyethylenimine; GalNAc = N-acetyl-galactosamine liver-targeting conjugate; LNA = locked-nucleic-acid; ASO = antisense oligo; GEMM = genetically engineered mouse model; CAM = chorio-allantoic membrane assay; ↑ = upregulated; ↓ = downregulated; → = results.

#### 3.4.3. Mechanistic Pathways Modulated by Therapeutic miRNAs

Whether restoring a tumor-suppressor miRNA or inhibiting an oncomiR, the ultimate goal is to tip the balance of cellular pathways from a pro-tumor to an anti-tumor state. MiRNA therapies accomplish this by altering the expression of numerous target genes in pathways governing cell survival, proliferation, angiogenesis, and immune surveillance. An unifying outcome of many successful miRNA interventions in HCC is the induction of apoptosis in cancer cells. For example, enforced miR-34a or miR-15/16 expression drives apoptosis by directly downregulating anti-apoptotic proteins like Bcl-2 and Mcl-1 [106,107,108]. Conversely, anti-miR treatments (miR-21, miR-221 inhibitors) relieve the repression of pro-apoptotic factors (e.g., PDCD4, Bim) and cell cycle checkpoints, triggering programmed cell death in tumor tissue [72,115,117,119,127]. Another major mechanism is cell cycle arrest: tumor-suppressor miRNAs such as miR-26a, -34a, -122, and -195 converge on halting the cell cycle in G_1_. These miRNAs target positive cell cycle regulators—cyclins (A, D1, E2) and cyclin-dependent kinases—as well as E2F transcription factors required for S-phase entry [83,84,105,107,108,126]. The result is an accumulation of HCC cells at the G_0_/G_1_ checkpoint and reduced proliferation. For instance, miR-26a replacement in HCC was shown to restore control of Cyclin D/E and induce a clear G_1_ arrest in vivo, and anti-miR-221 allows p27^Kip1^ to resume its cell cycle braking function, likewise leading to G_1_–S blockade [83,84,117,119]. By reinstating these internal “brakes”, miRNA therapies restrain cancer cells’ cell cycle, making them more susceptible to senescence or death.

Angiogenesis is another cancer hallmark affected by miRNAs. Some tumor-suppressor miRNAs inherently have anti-angiogenic activity. miR-195/15a/16 family mimics, for example, directly target the pro-angiogenic factor VEGFA, as well as downstream effectors like VAV2 and CDC42, thereby impairing the ability of HCC to develop new vasculature [108]. In HCC models, restoring miR-195 led to decreased microvessel density in tumors in conjunction with tumor growth inhibition [108]. Conversely, oncogenic miR-21 and miR-221 promote angiogenesis (partly by upregulating HIF1α and ANG2 via PTEN suppression); thus, their inhibition can indirectly exert an anti-angiogenic effect [115,117]. Indeed, anti-miR-21 has been found to reduce VEGF levels and tumor vascularization in some studies of liver cancer and chronic liver disease, suggesting improved oxygenation and drug delivery in the treated tumor [115]. Additionally, miRNA therapies that target angiogenesis complement their cell-intrinsic killing effects by starving the tumor of nutrients and oxygen.

Perhaps the most cutting-edge aspect of miRNA therapeutics lies in immune modulation. miRNAs are now known to shape the tumor immune microenvironment by acting within cancer cells and immune cells alike [4]. For instance, HCC-derived exosomal miR-21 and miR-452 can induce tumor-promoting M2 polarization of macrophages, aiding immune escape [44,115]. Therefore, anti-miR-21 therapy might not only act on tumor cells but also re-polarize tumor-associated macrophages towards an anti-tumor (M1) phenotype by preventing miR-21’s suppression of STAT1/NF-κB signaling in those immune cells [115]. Likewise, delivering miR-99b or miR-144/451 to macrophages has been shown experimentally to drive them into an M1 state and inhibit HCC growth via enhanced phagocytosis and antigen presentation [40,128]. MiRNA therapies can also influence cancer cells’ expression of immune checkpoint molecules. Restoring miR-34a in tumors was reported to downregulate PD-L1 on HCC cells, potentially increasing their visibility to cytotoxic T cells [106,107]. The AAV-miR-22 gene therapy described above provided a striking example of concomitant immune modulation: miR-22-treated tumors exhibited reduced infiltration of immunosuppressive cell types and an increase in activated T cells, correlating with superior tumor control [111]. Thus, beyond cell-autonomous tumor suppression, miRNA-based interventions can boost anti-tumor immunity—an effect that could be leveraged alongside immunotherapies. In summary, by modulating pathways of apoptosis, cell cycle, angiogenesis, and immune response, miRNA therapeutics attack HCC on multiple fronts, addressing both the cancer cells and their supportive microenvironment.

#### 3.4.4. Preclinical Models for miRNA Studies in HCC

Preclinical models for miRNA-focused hepatocellular carcinoma research span reductionist in vitro platforms to patient-derived in vivo systems. Canonical HCC cell lines (e.g., HepG2, Huh7) and 3D spheroid/organoid cultures enable rapid miRNA perturbation with readouts of proliferation, apoptosis, and metabolic flux, yet they incompletely capture intratumoral heterogeneity and the stromal–immune milieu [129,130]. Among in vivo approaches, subcutaneous xenografts facilitate pharmacokinetic/pharmacodynamic sampling but distort hepatic cues; orthotopic liver implantation better reproduces sinusoidal blood flow, hypoxia, and metastatic routes, and is therefore preferred for evaluating delivery strategies (e.g., lipid nanoparticles, viral vectors) and rational combinations (see Table 4) [131,132,133,134,135]. Etiology-relevant carcinogen or diet-induced models (e.g., DEN, CCl_4_/Western diet) recapitulate the inflammation–fibrosis–tumor sequence and enable longitudinal assessment of circulating miRNA biomarkers and immunometabolic reprogramming under therapy [136,137,138,139]. Transgenic and other GEMMs—whether oncogene-driven or miRNA-engineered (e.g., Myc; liver-specific miR-221 overexpression)—support in vivo validation of target engagement for mimic replacement (miR-26a/199a/195) or antagomir inhibition (miR-221/21) [103,140,141,142]. Finally, patient-derived xenografts preserve genetic and phenotypic heterogeneity and histopathology, more faithfully mirroring clinical drug response than cell-line models, and thus provide a robust platform for biomarker qualification and co-clinical testing of miRNA therapeutics, including mimic/antagomir regimens and ADC/immuno-oncology combinations [115,143,144]. Model selection should be aligned with the biological question, delivery modality, and planned clinical endpoints.

#### 3.4.5. Delivery Strategies and Clinical Translation

A central challenge in developing miRNA-based therapy for HCC is achieving efficient and safe delivery of the therapeutic RNA to tumor cells. Early studies employed viral vectors or lipid nanoparticles to introduce miRNA mimics/inhibitors into liver tumors, each with advantages and drawbacks. Viral delivery (using hepatotropic viruses like AAV or adenovirus) ensures high transduction efficiency and sustained miRNA expression, as demonstrated by the AAV-mediated miR-26a and miR-22 therapies that achieved robust tumor suppression in mice [103,111]. However, viral methods have dosage limits and potential safety concerns (insertional mutagenesis or immunogenicity) [103,111]. Non-viral carriers such as lipid nanoparticles (LNPs) and liposomes have therefore been at the forefront of miRNA drug development. LNPs can encapsulate synthetic miRNA mimics or anti-miRs and protect them from degradation while facilitating uptake into the liver. MRX34’s formulation exemplified this approach, successfully delivering miR-34a mimic into patient tumors and normal tissues [105,106]. The need for repeated high dosing of MRX34, however, revealed an immunostimulatory risk inherent to some double-stranded RNA mimics: several patients experienced cytokine release and immune-related toxicities attributed in part to the innate immune sensing of the miR-34a duplex [107]. Subsequent analyses suggested that the design of the mimic (double-stranded with certain subsequent ends) and the relatively large nanoparticle dose contributed to these effects [105]. In response, newer delivery strategies are exploring chemically stabilized single-stranded mimics and improved lipid carriers to minimize immune activation. The partial clinical success of an miR-16 mimic (TargomiR) delivered in bacterial minicells in another cancer indicates that delivery vehicles can greatly influence safety; TargomiR, despite using a double-stranded mimic, did not elicit severe immune reactions, pointing to the importance of the carrier and dosing schedule [145].

A 2024 systematic review of 31 randomized trials (*n* ≈ 10,399) of nanoparticle-based cancer therapies reported that, overall, nanoparticle arms did not demonstrate superiority in PFS/OS/pCR versus controls, with hematologic toxicities (lympho/leucopenia, neutropenia) most common; most studies evaluated paclitaxel-based platforms. These findings underscore that delivery technology alone is insufficient; payload and biology remain decisive, supporting our emphasis on mechanism-anchored miRNA therapeutics and careful clinical endpoint selection [146].

For oncomiR inhibitors, antisense oligonucleotides (ASOs) with chemical modifications (2′-O-methyl, LNA, phosphorothioate backbones, etc.) are commonly used to enhance stability and targeting. These ASOs can be conjugated to cholesterol or GalNAc to boost liver uptake. For example, a cholesteryl-modified anti-miR-21 was effective in mouse HCC models [115], and an LNA-based anti-miR-221 showed tumor uptake and long-lasting suppression of miR-221 in vivo [142]. Such chemistries have already been employed in human trials for other liver-related miRNA targets (notably anti-miR-122 miravirsen in hepatitis C patients, which achieved virus suppression by sequestering miR-122 [147]). Miravirsen’s success in safely modulating miR-122 in humans provides optimism that similar ASO approaches could be translated to HCC, with the caveat that in HCC, the objective would be restoring miR-122 (hence a different strategy) [147].

Another innovative platform uses extracellular vesicles (exosomes) as natural nanoparticles to ferry miRNA cargos. Exosomes exhibit inherent biocompatibility and liver tropism (especially if derived from liver cells or engineered with targeting ligands). Preliminary research has shown that exosomes loaded with tumor-suppressor miRNA mimics can be taken up by HCC cells and mediate gene silencing, although clinical data are not yet available [4]. Additionally, loco-regional delivery techniques are being explored: ultrasound-targeted microbubble destruction has been used to locally propel miRNA-loaded nanoparticles into HCC xenografts, achieving higher intratumoral concentrations and therapeutic effect while limiting systemic exposure [126]. This method was used to co-deliver miR-122 and anti-miR-21 to liver tumors in a rat model, resulting in improved efficacy over systemic delivery [126].

Despite encouraging preclinical results, clinical translation of miRNA therapies for HCC remains in early stages. The first-in-human trial with MRX34 (miR-34a mimic) provided valuable lessons; while it did not reach Phase II due to immune-related adverse events, the trial confirmed target engagement (with reduced levels of multiple oncogenes in patients’ white blood cells and tumors) and produced signs of anti-tumor activity [106]. These outcomes justify continued development of miR-34a-based therapeutics with improved designs—for instance, next-generation miR-34 mimics or mimetics that avoid triggering innate immunity. Meanwhile, no other miRNA mimic than MRX34 has entered advanced clinical trials for HCC until now, but several are in preclinical pipelines. miR-122 mimic therapy for HCC is one potential candidate given the strong rationale and the liver-specific nature of miR-122; while researchers are investigating safe vectors to reintroduce miR-122 in HCC without affecting normal liver function [148]. Recent RCT meta-evidence indicates that nanoparticle formulations alone have not consistently improved PFS/OS across cancers; thus, miRNA therapeutics should prioritize biologically decisive payloads with clear pharmacodynamic markers rather than rely solely on delivery tech [146].

Nevertheless, companies have developed anti-miR oligonucleotides against oncomiRs like miR-21 and miR-221, some of which have undergone early clinical trials in other cancers (e.g., anti-miR-21 for renal fibrosis and anti-miR-155 for lymphoma). It is plausible that as these molecules prove safe, they could be redirected to HCC, especially for patients who relapse on standard therapies [149,150]. Looking forward, the field is exploring combination strategies where miRNA therapeutics are used alongside conventional treatments. For example, a patient with advanced HCC might receive an anti-miR-221 agent to prime the tumor by lifting resistance mechanisms, followed by a tyrosine kinase inhibitor or immunotherapy to kill the now-vulnerable cancer cells [151,152]. Because miRNAs broadly reprogram cellular networks, they might prevent or delay the emergence of drug resistance—an effect highly desirable in HCC, where single-agent therapies often lose effectiveness over time.

## 4. Challenges and Future Perspectives

Comparative studies in animals—such as an miR-199a-3p mimic rivaling sorafenib, or miR-22 gene therapy outperforming lenvatinib [104,123]—raise hope that miRNA interventions could complement or even challenge conventional HCC therapies in efficacy. Importantly, miRNA therapies may work synergistically with existing treatments: for instance, combining a multi-targeted miRNA with an immune checkpoint inhibitor could simultaneously kill tumor cells and invigorate the immune response, a two-pronged attack difficult to achieve with standard drugs alone. As research progresses, patient stratification will also be key—identifying which miRNA dysregulations drive a given patient’s tumor will allow selection of tailored miRNA therapeutics, aligning with the goals of precision oncology. In summary, the therapeutic targeting of miRNAs in HCC has evolved from a compelling concept to a growing body of evidence, and it holds the potential to unlock new, more effective treatment paradigms for this challenging liver cancer. With continued advances, miRNA-based therapies could in the future become an integral part of the HCC treatment landscape, used in combination with surgery, kinase inhibitors, and immunotherapies to improve patient outcomes.

The challenges are that circulating miRNA studies in HCC are hampered by pre-analytical and analytical variability: differences in sample type (serum vs. plasma), collection and processing protocols, RNA extraction kits, and quantification platforms (qRT-PCR, microarray, sequencing) can yield non-biological fluctuations in measured levels. Because many candidate miRNAs are present at low abundance, even minor procedural inconsistencies—hemolysis, freeze–thaw cycles, or choice of normalization control—can distort results and hinder cross-study comparisons.

Beyond technical issues, biological heterogeneity poses a formidable barrier. HCC arises in the context of diverse underlying liver diseases (HBV, HCV, alcohol, NAFLD/NASH), each of which influences baseline hepatic and circulating miRNA patterns. Geographic and ethnic differences further affect tumor biology and microenvironmental cues, meaning that a miRNA signature validated in one population may not generalize elsewhere. Many published cohorts are single-center and retrospective with modest sample sizes, raising the risk of overfitting multi-miRNA panels or missing confounding factors such as co-morbidities and treatment histories.

Translating mechanistic insights into safe, effective miRNA-based therapies remains an ongoing challenge. Delivery methods—viral vectors, lipid nanoparticles, GalNAc conjugates—must achieve sufficient on-target concentrations in tumor or liver tissue without provoking off-target effects or immune activation. The early MRX34 trial illustrated both promise and peril: while tumor delivery and target engagement were demonstrable, immune-related toxicities halted further development. Iterative improvements in oligonucleotide chemistry, single-strand designs, and targeted delivery platforms will be essential, as will robust preclinical safety studies and small-scale human trials.

Finally, as a narrative rather than a systematic review, this work reflects the authors’ selection of illustrative studies rather than an exhaustive, protocol-driven appraisal. While this approach allows integration of mechanistic depth with clinical context, it may overlook some niche findings or unpublished null results. Readers should interpret the presented miRNA candidates and strategies as representative of major trends, with the understanding that ongoing and future studies may expand, refine, or revise these paradigms.

## 5. Conclusions

In conclusion, miRNA-based therapies represent a novel and comprehensive approach to HCC treatment, targeting the disease at the post-transcriptional regulatory level. Both miRNA replacement (for tumor suppressors) and miRNA inhibition (for oncomiRs) have demonstrated potent anti-HCC effects in models, impacting pathways of cell cycle, apoptosis, angiogenesis, and immune evasion that underlie tumor growth. While no miRNA therapeutics have yet reached approval, the ongoing refinements in delivery systems and safety profiles are addressing the initial setbacks.

## Figures and Tables

**Figure 1 biomedicines-13-02243-f001:**
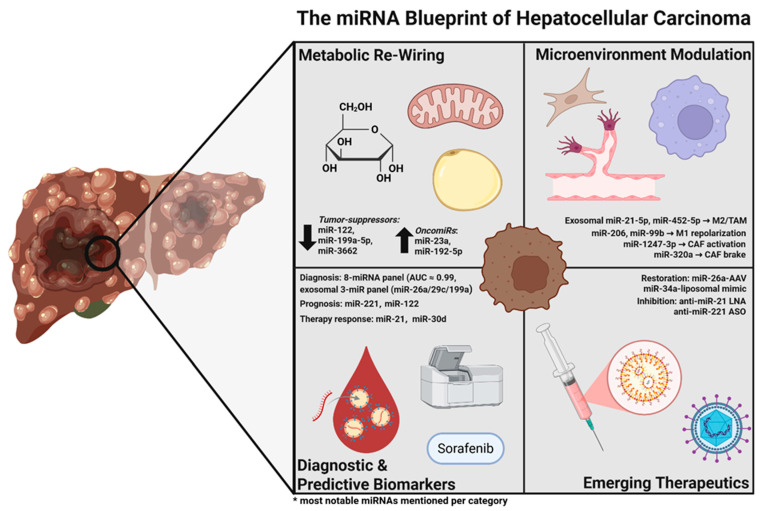
The miRNA Blueprint of Hepatocellular Carcinoma. Schematic overview of the multifaceted roles of microRNAs in hepatocellular carcinoma (HCC). The graphic is partitioned into four panels: 1. Metabolic Re-Wiring: Loss of tumor-suppressor miRNAs (e.g., miR-122, miR-199a-5p, miR-3662) and gain of oncomiRs (miR-23a, miR-192-5p) remodel glucose, lipid, and glutamine metabolism to favor the Warburg phenotype and fatty-acid oxidation. 2. Microenvironment Modulation: Tumor-derived exosomal miRNAs (miR-21-5p, miR-452-5p, miR-1247-3p) skew macrophages to an M2/TAM phenotype and activate CAFs; conversely, miR-206/miR-99b repolarize TAMs to M1, while miR-138-5p and miR-101 restrain angiogenesis and vascular mimicry. 3. Diagnostic and Predictive Biomarkers: Circulating and exosomal miRNAs (single markers such as miR-21, miR-122, or multi-miRNA panels) achieve AUCs up to ~0.99 for early HCC detection, differential diagnosis vs. cirrhosis, and prognostication (miR-221, miR-30d) or therapy response (miR-21, miR-486-3p). 4. Emerging Therapeutics: Replacement of tumor-suppressor miRNAs (miR-26a-AAV, miR-34a-liposome, miR-199a-3p mimics) and inhibition of oncomiRs (anti-miR-21 LNA, anti-miR-221 ASO) delivered via viral vectors, lipid nanoparticles, GalNAc-ASO, or ultrasound-microbubble platforms. Only the most notable miRNAs are shown in each category; arrows indicate up- or down-regulation in HCC. AAV = adeno-associated virus; ASO = antisense oligonucleotide; CAF = cancer-associated fibroblast; TAM = tumor-associated macrophage. “*” denotes most notable miRNAs mentioned in all four categories in the figure.

**Table 1 biomedicines-13-02243-t001:** MicroRNAs implicated in cancer cell metabolism and TME remodeling in HCC. MicroRNAs that coordinate metabolic rewiring within HCC cells and orchestrate stromal, immune, and angiogenic changes in the tumor micro-environment. miRNAs are grouped into intrinsic metabolic regulators (A) and intercellular mediators of micro-environmental crosstalk (B).

miRNA	Primary Target(s)	Pathway/Cell Type Affected	Net Effect on HCC Biology	Dys-Reg.	Mode of Action	Ref.
**A. Cancer-Cell-Intrinsic Metabolic Regulation**						
**Glycolysis and Glutaminolysis**						
**miR-122**	PKM2, GLS1, SLC1A5/ASCT2, G6PD	Warburg glycolysis, glutaminolysis, PPP in tumor cells	Restores oxidative metabolism; ↓ lactate and glutamine use → growth restraint	↓	TS	[31,33]
**miR-3662**	HIF-1α	HIF-driven glycolytic programming	↓ GLUT1/HK2/PKM2/LDHA → curtailed glycolysis and tumor growth	↓	TS	[15,16]
**miR-199a-5p**	HIF-1α	Warburg glycolysis	↓ glucose uptake and lactate → slower proliferation	↓	TS	[19,56]
**miR-885-5p**	HK2	Glycolysis	Blunts aerobic glycolysis under hypoxia	↓	TS	[17]
**miR-125a**	HK2	Glycolysis and ROS balance	↓ glucose consumption, lactate and ROS	↓	TS	[18,40]
**miR-192-5p**	c-Myc axis (GLUT1, PFKFB3)	Glycolysis	Loss fuels glycolysis; re-expression dampens invasiveness	↓	TS	[20]
**miR-23a**	PPARGC1A/PGC-1α, G6PC	Gluconeogenesis ↔ glycolysis switch	IL-6/STAT3-induced Warburg enhancement	↑	ONC	[22]
**Lipid Metabolism and β-Oxidation**						
**miR-148a**	c-Myc, DNMT1, PGC-1α, SIRT7	Lipogenesis, FAO, OXPHOS	Restoring curbs lipid accumulation and tumor growth		TS	[32]
**miR-4310**	FASN, SCD1	De novo fatty-acid synthesis	↓ lipogenesis → ↓ proliferation/metastasis	↓	TS	[23]
**miR-377-3p**	CPT1C	Mitochondrial FA β-oxidation	Impairs FA import; suppresses growth and metastasis	↓	TS	[24]
**miR-612**	HADHA	Terminal FA β-oxidation	Re-expression limits metastasis; low levels mark aggressiveness	↓	TS	[15,25]
**Amino-Acid Transport/Glutamine Axis**						
**miR-137**	SLC1A5/ASCT2	Glutamine uptake and anaplerosis	Restored miR-137 ↓ glutamine flux → tumor inhibition	↓	TS	[34]
**B. Tumor-Micro-Environment Modulation**						
**Immune-Cell Reprogramming (TAM-centric)**						
**miR-21-5p**	RhoB (also PTEN)	Macrophage M2 polarization	Immunosuppression; supports growth and poor prognosis	↑	ONC	[38]
**miR-452-5p**	TIMP3	TAM M2 shift, ECM remodeling	↑ M2 TAMs and metastasis	↑	ONC	[44]
**miR-23a-3p**	PTEN	PI3K/AKT → PD-L1 on TAMs	T-cell suppression/immune escape	↑	ONC	[22]
**miR-206**	KLF4/NF-κB axis	Macrophage M1 activation	↑ CD8^+^ T-cell recruitment; anti-tumor immunity	↓	TS	[39]
**miR-99b**	κB-Ras2, mTOR, IRF4	TAM re-programming to M1	↑ phagocytosis and antigen presentation	↓	TS	[40]
**Fibroblast/CAF Activation**						
**miR-1307-3p**	DAB2IP	Hypoxia-AKT/mTOR loop in CAF-like milieu	Promotes survival, invasion and HIF-1α feed-back	↑	ONC	[35]
**miR-1247-3p**	B4GALT3	β1-integrin/NF-κB in fibroblasts	Converts fibroblasts to IL-6/8-secreting CAFs → EMT/metastasis	↑	ONC	[43,50]
**miR-130b-3p**	HOXA5	PI3K/AKT/mTOR and VEGF	Drives angiogenic CAF phenotype	↑	ONC	[46]
**miR-320a**	PBX3	MAPK signaling in HCC cells	Paradoxical growth restraint; tumor-suppressive	↓	TS	[45]
**miR-101**	TGFβR1, SMAD2 (HCC); SDF1 (CAF)	CAF-induced vascular mimicry	Blocks VM and neovascularization	↓	TS	[54]
**Angiogenesis and Vascular Permeability**						
**miR-210**	SMAD4, STAT6	Hypoxia-driven angiogenesis (endothelium)	Abnormal vessel formation; pro-tumor	↑	ONC	[29,47]
**miR-103**	VE-cadherin, p120-catenin, ZO-1	Endothelial junction integrity	↑ vascular permeability → intravasation	↑	ONC	[51]

TS = tumor-suppressor miRNA, ONC = oncogenic (pro-tumor) miRNA. ↑ = upregulated, ↓ = downregulated, → = results.

**Table 2 biomedicines-13-02243-t002:** Selected miRNAs and miRNA panels proposed as diagnostic biomarkers in HCC. Individual and multiplex circulating or exosomal miRNAs that differentiate hepatocellular carcinoma from healthy, cirrhotic, or chronic hepatitis controls, listing sample source, diagnostic setting, ROC performance dysregulation direction, and key references.

miRNA	Sample Source	Comparison Group(s)	Diagnostic Context	Performance (AUC, Sens, Spec)	Dysregulation in HCC	Ref.
**miR-21 (single)**	Serum	Healthy vs. HCC	General HCC detection (all stages)	AUC 0.849, 82% sens, 84% spec vs. healthy, also AUC ~0.81 vs. LC	↑	[60]
**miR-122 (single)**	Serum	Chronic hepatitis vs. HCC	HCC vs. chronic liver disease	AUC 0.892 (95% CI 0.84–0.93) vs. chronic hepatitis	↓	[58]
**miR-221 (single)**	Serum	Chronic hepatitis vs. HCC	HCC vs. chronic liver disease	AUC ~0.806 (95% CI 0.75–0.86) vs. chronic hepatitis	↑	[58]
**miR-1246 (single)**	Plasma	Cirrhotic/healthy vs. HCC	HCC detection (mixed controls)	AUC ~0.812 in meta-analysis; elevated in HCC plasma vs. both cirrhosis and healthy	↑	[58]
**miR-26a (single)**	Serum	Chronic hepatitis vs. HCC	HCC vs. chronic liver disease	AUC ~0.867 (95% CI 0.81–0.91) vs. chronic hepatitis	↓	[58]
**8-miRNA panel (e.g., Yamamoto et al.)**	Serum	At-risk (LC + CH) vs. HCC	Early HCC detection (Stage I/II)	AUC 0.99, 98% sensitivity for Stage I HCC	(panel of ↑/↓)	[64]
**3-miRNA** **exosomalpanel** **(miR-26a/29c/199a)**	Exosomes (plasma)	LC and healthy vs. HCC	Early detection and differential	AUC 0.994 (100% sens, 96% spec) vs. healthy; 0.965 vs. LC	↓	[28]
**5-miRNA EV panel (miR-183/19a/148b/34a/215)**	Extracellular vesicles	Non-HCC controls vs. HCC	General HCC detection	~90% sensitivity, 92% specificity in mixed cohort	↑	[65]
**miR-221 + miR-29c (two-miRNA combo)**	Serum	Healthy vs. early HCC	Early HCC vs. non-cancer	AUC ~0.97 for Stage I–II HCC vs. normal; detected ~85% of early HCC vs. 46% by AFP	↑ (miR-221); ↓ (miR-29c)	[59]
**3-miRNA plasma panel (miR-126/206/222) + AFP**	Plasma + AFP	Healthy vs. HCC	HCC detection (all stages)	AFP alone AUC 0.889; combined panel + AFP AUC 0.989 (sens/spec ~97%/98%)	(panel of ↑/↓)	[71]

LC = liver cirrhosis; CH = chronic hepatitis (unspecified); AFP = alpha-fetoprotein; Dysregulation indicates direction of change in HCC (↑ upregulated in HCC, ↓ downregulated in HCC).

**Table 3 biomedicines-13-02243-t003:** Predictive miRNAs in HCC: targets, context, predictive value, dysregulation, and key references. MicroRNAs with validated predictive value in hepatocellular carcinoma, organized by clinical scenario. (A) lists miRNAs that forecast response or resistance to systemic therapy (chiefly sorafenib); (B) covers predictors of locoregional efficacy after trans-arterial chemoembolisation (TACE); (C) details tissue-, blood- and exosome-derived miRNAs that anticipate tumor recurrence following ostensibly curative surgery or ablation; and (D) summarises single-miRNA and composite signatures that stratify overall or disease-free survival beyond standard staging.

miRNA	Target(s)/Function	Context	Predictive Value	Dysregulation (High/Low)	Ref.
**A. Systemic Therapy Response**					
**miR-21**	PTEN (tumor suppressor)	Sorafenib response	Sorafenib resistance marker	↑ in resistant cells	[75]
**miR-30d**	—(secreted biomarker)	Sorafenib response	Sorafenib responder marker	↑ in responders	[68,76]
**miR-486-3p**	FGFR4, EGFR (oncogenes)	Sorafenib resistance	Sorafenib resistance marker	↓ in resistant tumors	[77]
**miR-25**	FBXW7 (autophagy regulator)	Sorafenib resistance	Sorafenib resistance marker	↑ in resistant tumors	[97]
**miR-423-5p**	Autophagy-related genes	Sorafenib response	Sorafenib response biomarker	↑ in resistant tumors	[98]
**B. Locoregional-Therapy Response**					
**miR-21**	PTEN	TACE	Early TACE failure	↑ in refractoriness	[84]
**miR-26a**	Cyclin D2/E2	TACE	Early TACE failure	↑ in refractoriness	[84]
**miR-29a-3p**	(DNMT-related)	TACE	Early TACE failure	↑ in refractoriness	[84]
**miR-1271**	(tumor-suppressive)	TACE response	Poor TACE response marker	↓ in non-responders	[80]
**miR-214**	(tumor-suppressive)	TACE response	Poor TACE response marker	↓ in non-responders	[99]
**miR-133b**	(tumor-suppressive)	TACE response	Poor TACE response marker	↓ in non-responders	[83]
**miR-335**	(tumor-suppressive)	TACE response	Poor TACE response marker	↓ in non-responders	[100]
**C. Recurrence after Curative Therapy**					
**miR-122**	Cyclin G1, ADAM17 (oncogenes)	Post-resection	↑ recurrence risk (↓ RFS)	↓ in tumors	[86]
**miR-15b**	Bcl-w (anti-apoptotic)	Post-resection	↓ recurrence risk (↑ RFS)	↑ in non-recurrers	[87]
**miR-34a**	Bcl-2, Cyclins	Post-ablation (RFA)	Early recurrence marker	↓ in recurrences	[88]
**miR-483-3p**	IGF2 locus, multiple	Post-resection (advanced HCC)	Recurrence predictor	↓ in recurrences	[89]
**miR-3201**	—(biomarker candidate)	Curative therapy (resection/RFA)	Responder vs. non-responder	↓ in complete responders	[90]
**miR-215-5p**	(exosomal oncogenic signals)	Curative therapy (exosomal, serum)	Shorter DFS (poor prognosis)	↑ in recurrences	[96]
**miR-92b**	(exosomal oncogenic signals)	Post-surgery (exosomal, serum)	Recurrence marker	↑ in recurrences	[91]
**D. Overall/Disease-Free Survival**					
**miR-221**	PTEN, CDKN1B (tumor suppressors)	General prognosis	Poor OS/DFS (high risk)	↑ in poor outcome	[94]
**3-miR signature (miR-139-3p, miR-760, miR-7-5p)**	(various tumor suppressors)	General prognosis (TCGA signature)	Risk score for OS	↓ (low-risk) or ↑ (high-risk)	[95]

↑ upregulated, ↓ downregulated, OS overall survival, DFS disease-free survival, RFS recurrence-free survival, RFA radiofrequency ablation, TACE trans-arterial chemoembolisation, tx treatment, pts patients.

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
