# Peer review of "MicroRNA Landscape in Hepatocellular Carcinoma: Metabolic Re-Wiring, Predictive and Diagnostic Biomarkers, and Emerging Therapeutic Targets"

_biomedicines, 2025, doi:10.3390/biomedicines13092243_

Round 1

Reviewer 1 Report

Comments and Suggestions for Authors

This article primarily discusses the advancements in research pertaining to microRNAs associated with hepatocellular carcinoma (HCC). It elucidates the role of miRNAs in regulating tumor glycolysis, lipid metabolism, glutamine metabolism, and immune remodeling within the tumor microenvironment. The paper further delves into the significance of miRNAs in early HCC diagnosis and their potential to predict resistance and therapeutic outcomes. Additionally, it introduces methodologies to restore tumor-suppressive miRNAs through mimics or AAV vectors, and to inhibit oncogenic miRNAs using antagomirs or LNA oligonucleotides, along with their anti-tumor effects in model systems. It also analyzes the challenges encountered by current technologies and encapsulates the accomplishments in target binding and safety optimization from preliminary trials. Conclusively, it suggests integrating miRNA biomarkers into monitoring algorithms and amalgamating miRNA therapies with existing treatments for precise management of HCC. The author is advised to revise the manuscript in accordance with the following comments.

  1. The author is advised to allocate additional space in the introduction section to provide comprehensive background information on hepatocellular carcinoma and microRNA.
  2. As a comprehensive and scholarly review, it should incorporate the most recent and authoritative references. It has been observed that several key references are missing from the current version. The following sources are closely related to the author's topic and are therefore recommended for inclusion to enhance the credibility and depth of the review.

https://doi.org/10.1016/j.ntm.2024.100036

https://doi.org/10.1002/med4.27

https://doi.org/10.1002/INMD.20250016

  1. The author is encouraged to provide a more comprehensive analysis of the metabolic regulatory mechanisms, including detailed information on specific target genes and relevant signaling pathways.
  2. Please perform a comprehensive analysis of the variations in specific miRNAs across different etiological factors.
  3. There are specific sections pertaining to animal models and clinical applications that necessitate a more comprehensive discussion to further elucidate their significance.

Author Response

September 04, 2025

Biomedicines

RE: Submission of REVISED manuscript (biomedicines-3835441)

Dear Reviewer

Please find enclosed our REVISED manuscript entitled “MicroRNA Landscape in Hepatocellular Carcinoma: Metabolic Re-Wiring, Predictive & Diagnostic Biomarkers, and Emerging Therapeutic Targets” to be considered for publication. We would like to thank you and the other reviewers for your thoughtful evaluation of our manuscript and your most welcome comments/suggestions. Accordingly, we have now revised our manuscript thoroughly to reflect these comments.

Please find below a point-by-point response to ALL the issues raised by you:

Reviewer 1

Comment 1: The author is advised to allocate additional space in the introduction section to provide comprehensive background information on hepatocellular carcinoma and microRNA.

Response 1: We thank the reviewer for this helpful suggestion. We expanded the Introduction to (i) provide concise epidemiologic context, risk factors, and current standards for HCC diagnosis/surveillance [page 2, paragraph 1, line 56-63], and (ii) add a short primer on miRNA biogenesis, stability, and measurement platforms, motivating their translational relevance in HCC [page 2, paragraph 3, line 85-90].

Comment 2: As a comprehensive and scholarly review, it should incorporate the most recent and authoritative references. It has been observed that several key references are missing from the current version. The following sources are closely related to the author's topic and are therefore recommended for inclusion to enhance the credibility and depth of the review.

https://doi.org/10.1016/j.ntm.2024.100036

https://doi.org/10.1002/med4.27

https://doi.org/10.1002/INMD.20250016

Response 2: We appreciate these timely recommendations and have integrated them where most relevant. Ma 2024 is cited in Section 3.4.5 Delivery Strategies and Clinical Translation to contextualize nanomedicine RCT evidence [page 22, paragraph 3, line 725-731]. Huang 2023 is cited in Section 3.1.1 Inter-Cellular Wiring to acknowledge neuro-tumor signaling within the tumor microenvironment [page 6, paragraph 2, line 223-226]. Li 2025 is cited in a new Section 3.4.4 Preclinical Models created after the review of comment 5 [page 22, paragraph 1, line 694-698].

Comment 3: The author is encouraged to provide a more comprehensive analysis of the metabolic regulatory mechanisms, including detailed information on specific target genes and relevant signaling pathways.

Response 3: Thank you. We expanded Section 3.1 miRNA-Driven Metabolic Effects in HCC with two new mechanism-focused paragraphs itemizing direct targets (e.g., HK2, PFKFB3, HIF1A, PKM2, G6PD, SLC1A5/ASCT2, GLS, CPT1C, HADHA) and mapped them onto PI3K/AKT/mTOR, HIF-1α/c-Myc, AMPK/PGC-1α, and TGF-β axes [page 5, paragraph 2, line 182-194].

Comment 4: Please perform a comprehensive analysis of the variations in specific miRNAs across different etiological factors.

Response 4: Thank you for highlighting this important point. We agree and have added a concise discussion in the Diagnostics section that contextualizes biomarker performance by etiology (HBV, HCV, NAFLD/MASLD, alcohol) and geography, with notes on pre-analytical standardization and the need for stratified validation. [page 10, paragraph 2-4, line 334-356].

Comment 5: There are specific sections pertaining to animal models and clinical applications that necessitate a more comprehensive discussion to further elucidate their significance.

Response 5: We created a new Section 3.4.4 Preclinical Models for miRNA Studies in HCC (covering 2D/3D, xenograft/orthotopic, chemical, transgenic, and PDX with pros/cons) and expanded Section 3.14 Delivery Strategies and Clinical Translation with pragmatic clinical-trial and implementation considerations (assay standardization, regulatory aspects, and lessons from nanomedicine RCTs) [page 21, paragraph 3, line 678-699].

Trusting that we have adequately addressed the reviewers' concerns, we would like to thank you for your help in improving our work significantly.

Kind regards,

Koustas Evangelos, MD, PhD

Reviewer 2 Report

Comments and Suggestions for Authors

In this review article, Liapopoulos et al aimed to summarize existing literature on miRNA regulation of hepatocellular carcinoma. This is a very comprehensive analysis, covering the breadth and depth of existing literature for the miRNA regulation of an aggressive tumor type. However, it is relatively convoluted and there are certain aspects that need to be addressed to be considered suitable for publication:

Major comments

1) I propose the Main Theme to be divided into the four core concepts of the article as follows:

3.1. miRNA-Driven Metabolic Effects in HCC

3.2 Diagnostic miRNAs in HCC

3.3 Predictive miRNAs in HCC

3.4 MiRNAs as Therapeutic Targets in HCC

The in-between sections can be re-numbered as 3.1.1, 3.1.2 etc. This organization will add clarity for the readers.

2) The manuscript is very densely written, and many sections require further explanation.

i) All genes and abbreviations should be defined the first time they appear in the manuscript. For example, line 149 "FA" are not defined but they are referred to as "fatty acids" in line 144.

ii) The authors should first briefly discuss the cellular function of certain pathways and axes (e.g. GLUT1/HK2/PKM2/LDHA) in order to then address the importance of their dysregulation by miRNAs. Schematics would greatly help with that.

iii) Please briefly define HCC-derived exosomes and their importance.

3) Certain concepts are mentioned in multiple parts of the manuscript and are not connected. For example, angiogenesis is mentioned with respect to tumor microenvironment under 3.2 and under 3.13. Another example would be AAV-delivered gene therapies 3.11, 3.13 and then technique limitations are mentioned under 3.14. Please carefully review the manuscript and if needed, re-organize the sections to ensure that concepts are fully analyzed in one place.

4) The reader would benefit from the description of miRNA atlases that are organ- and disease-specific.

Minor comments

5) There are 2 tables named as "Table 1". Please correct all the Tables accordingly.

6) Line 58: "On the order of" -> In the order of

7) Line 64: When the authors mention "biopsy", I assume they mean histopathological evaluation. Please revise.

8) Line 66: What is the importance of alpha-fetoprotein?

9) Line 194: "Including" is mentioned twice. Please remove.

10) Table 1: Please define the abbreviations "TS" and "ONC".

11) Line 222-238: "Several individual...multi-marker strategies[54]" can be a subsection with the title: Early Detection of HCC via Individual miRNA, to match the concept of the next subsection "Early Detection of HCC via miRNA Panels".

12) Line 241: "8-miRNA serum panel": Please define the miRNAs.

13) Line 259:"background liver disease". Do the authors mean concurrent non-malignant liver disease?

14) Line 276:"a background of cirrhosis". Please rephrase

15) Line 314: "is linked to acquired resistance". Please add therapeutic resistance

16) Line 351:"in patients with advanced histology". Please specify.

17) Lines 429-430. Please address the gap.

18) Line 537: Please remove "which is"

Author Response

September 04, 2025

Biomedicines

RE: Submission of REVISED manuscript (biomedicines-3835441)

Dear Reviewer

Please find enclosed our REVISED manuscript entitled “MicroRNA Landscape in Hepatocellular Carcinoma: Metabolic Re-Wiring, Predictive & Diagnostic Biomarkers, and Emerging Therapeutic Targets” to be considered for publication. We would like to thank you and the other reviewers for your thoughtful evaluation of our manuscript and your most welcome comments/suggestions. Accordingly, we have now revised our manuscript thoroughly to reflect these comments.

Please find below a point-by-point response to ALL the issues raised by you:

Reviewer 2

Comment 1: I propose the Main Theme to be divided into the four core concepts of the article as follows:

3.1. miRNA-Driven Metabolic Effects in HCC

3.2 Diagnostic miRNAs in HCC

3.3 Predictive miRNAs in HCC

3.4 MiRNAs as Therapeutic Targets in HCC

The in-between sections can be re-numbered as 3.1.1, 3.1.2 etc. This organization will add clarity for the readers.

Response 1: Thank you for this suggestion. We agree that the proposed framework will help readers navigate the review. Accordingly, we reorganized Section 3 into the four requested pillars and renumbered subordinate sections (3.1.1, 3.1.2, etc.).

Comment 2: The manuscript is very densely written, and many sections require further explanation.

  1. i) All genes and abbreviations should be defined the first time they appear in the manuscript. For example, line 149 "FA" are not defined but they are referred to as "fatty acids" in line 144.
  2. ii) The authors should first briefly discuss the cellular function of certain pathways and axes (e.g. GLUT1/HK2/PKM2/LDHA) in order to then address the importance of their dysregulation by miRNAs. Schematics would greatly help with that.

iii) Please briefly define HCC-derived exosomes and their importance.

Response 2:

  1. i) Thank you for flagging the need for clarity at first mention. We performed a global pass to ensure that all genes and abbreviations are defined on first use (e.g., fatty acids [FA]) [page 5, paragraph 1, line 178].
  2. ii) Thank you for proposing short pathway primers and additional schematics. We appreciate the reviewer’s suggestion. However, given the length and complexity of the current manuscript—and the fact that our target readership is already familiar with core glycolytic and lipid/glutamine pathways, we opted not to add additional pathway primers or new schematics to keep the paper focused and within word limits.

iii) Thank you for asking us to define exosomes. We now include a concise definition at their first mention, clarifying size range, biogenesis/relevance, and why exosomal miRNAs are attractive as stable liquid-biopsy analytes [page 9, paragraph 3, line 315-320].

Comment 3: Certain concepts are mentioned in multiple parts of the manuscript and are not connected. For example, angiogenesis is mentioned with respect to tumor microenvironment under 3.2 and under 3.13. Another example would be AAV-delivered gene therapies 3.11, 3.13 and then technique limitations are mentioned under 3.14. Please carefully review the manuscript and if needed, re-organize the sections to ensure that concepts are fully analyzed in one place.

Response 3: Thank you; we agree that avoiding redundancy is important. At the same time, we intentionally keep brief, context-specific recaps in sections that readers often consult independently (Diagnostics vs Therapeutics), so that each part remains self-contained.

Comment 4: The reader would benefit from the description of miRNA atlases that are organ- and disease-specific.

Response 4: Excellent point. We inserted a brief boxed overview (“miRNA resources and atlases relevant to HCC”) highlighting TCGA-LIHC, OncoMir/OncomiR, miRmine, miRGator, and exoRBase for our readers. [page 4, paragraph 1, line 136]

Comment 5: There are 2 tables named as "Table 1". Please correct all the Tables accordingly.

Response 5:Thank you for catching this. We standardized table numbering: Table 1 (metabolic/TME modulators), Table 2 (diagnostics), Table 3 (therapeutics), Table 4 (preclinical models), and updated all in-text citations.

Comment 6: Line 58: "On the order of" -> In the order of

Response 6: Good suggestion. We made the change. [page 2, paragraph 1, line 66]

Comment 7: Line 64: When the authors mention "biopsy", I assume they mean histopathological evaluation. Please revise.

Response 7: Agreed. We now specify “histopathological evaluation.” [page 2, paragraph 2, line 73]

Comment 8: Line 66: What is the importance of alpha-fetoprotein?

Response 8: Thank you for asking us to clarify this. We added a one-sentence explanation of AFP’s role in surveillance and its limitations for early-stage detection and specificity. [page 8, paragraph 1, line 265-269]

Comment 9: Line 194: "Including" is mentioned twice. Please remove.

Response 9: Agreed. We removed the duplication and smoothed the sentence. [page 6, paragraph 4, line 244]

Comment 10: Table 1: Please define the abbreviations "TS" and "ONC".

Response 10: Agreed. The table legend already states “TS = tumour-suppressor miRNA; ONC = oncogenic miRNA.” [page 8, paragraph 1, line 262-263]

Comment 11: Line 222-238: "Several individual...multi-marker strategies[54]" can be a subsection with the title: Early Detection of HCC via Individual miRNA, to match the concept of the next subsection "Early Detection of HCC via miRNA Panels".

Response 11: Great idea for symmetry. We created “Early Detection of HCC via Individual miRNAs” immediately before the existing panels subsection. [page 9, paragraph 1, line 277-295]

Comment 12: Line 241: "8-miRNA serum panel": Please define the miRNAs.

Response 12: Thanks for the prompt to be concrete. We now list the eight miRNAs. [page 9, paragraph 2, line 298-299]

Comment 13: Line 259:"background liver disease". Do the authors mean concurrent non-malignant liver disease?

Response 13: We agree the phrasing can be clearer. We revised to “concurrent non-malignant liver disease.” [page 9, paragraph 3, line 320]

Comment 14: Line 276:"a background of cirrhosis". Please rephrase

Response 14: Good wording improvement. We now use “in cirrhotic livers,” [page 10, paragraph 4, line 361]

Comment 15: Line 314: "is linked to acquired resistance". Please add therapeutic resistance

Response 15: Thank you for the precision. The text now reads “acquired therapeutic resistance.” [page 12, paragraph 1, line 399]

Comment 16: Line 351:"in patients with advanced histology". Please specify.

Response 16: Agreed. We replaced the phrase with “poorly differentiated (Edmondson–Steiner grade III–IV) and/or microvascular invasion,” which better reflects the clinical construct used. [page 13, paragraph 3, line 438-440]

Comment 17: Lines 429-430. Please address the gap.

Response 17: Thank you for the comment. Unfortunately we were unable to understand what “gap” referred to, and consequently couldn’t resolve this comment.

Comment 18: Line 537: Please remove "which is"

Response 18: Agreed. We deleted “which is”

Trusting that we have adequately addressed the reviewers' concerns, we would like to thank you for your help in improving our work significantly.

Kind regards,

Koustas Evangelos, MD, PhD

Reviewer 3 Report

Comments and Suggestions for Authors

This manuscript entitled "MicroRNA Landscape in Hepatocellular Carcinoma: Metabolic Re-Wiring, Predictive & Diagnostic Biomarkers, and Emerging Therapeutic Targets" presents a comprehensive and up-to-date narrative review on the multifaceted roles of microRNAs (miRNAs) in hepatocellular carcinoma (HCC). The paper effectively covers most of literature covering miRNA-driven metabolic reprogramming, circulating and exosomal miRNAs as diagnostic and predictive biomarkers, and miRNA-based therapeutic strategies, with particular focus on the translational potential in HCC management. SOme of the minor suggestions are below- 

  • Figure 1 legend could be more descriptive
  • Table 1 is mentioned 2 times. Authors must correct this with correct sequence and proper in text citation
  • some typographical mistakes - e.g. line 194-195 - "including" term is in duplicate.
  • Given geographic and etiologic variability in HCC, discussing how miRNA biomarkers perform across diverse populations and liver disease etiologies would be valuable.
  • A concise tabulation of miRNA-based therapeutics under clinical or preclinical development could benefit readers looking for a snapshot of pipeline drugs.

With minor revisions to enhance clarity, transparency, and completeness, this manuscript is suitable for publication in Biomedicines.

Author Response

September 04, 2025

Biomedicines

RE: Submission of REVISED manuscript (biomedicines-3835441)

Dear Reviewer

Please find enclosed our REVISED manuscript entitled “MicroRNA Landscape in Hepatocellular Carcinoma: Metabolic Re-Wiring, Predictive & Diagnostic Biomarkers, and Emerging Therapeutic Targets” to be considered for publication. We would like to thank you and the other reviewers for your thoughtful evaluation of our manuscript and your most welcome comments/suggestions. Accordingly, we have now revised our manuscript thoroughly to reflect these comments.

Please find below a point-by-point response to ALL the issues raised by you:

Reviewer 3

Comment 1: Figure 1 legend could be more descriptive

Response 1: Thank you for this helpful suggestion. We agree that a richer legend will aid readers who scan figures first. We have expanded the Figure 1 legend to explain each panel and define key abbreviations

Comment 2: Table 1 is mentioned 2 times. Authors must correct this with correct sequence and proper in text citation

Response 2: Thank you for catching this. We standardized table numbering: Table 1 (metabolic/TME modulators), Table 2 (diagnostics), Table 3 (therapeutics), Table 4 (preclinical models), and updated all in-text citations.

Comment 3: some typographical mistakes - e.g. line 194-195 - "including" term is in duplicate.

Response 3: Agreed. We removed the duplication and smoothed the sentence. [page 6, paragraph 4, line 244]. We performed a global pass to ensure other potential typographical mistakes were corrected.

Comment 4: Given geographic and etiologic variability in HCC, discussing how miRNA biomarkers perform across diverse populations and liver disease etiologies would be valuable.

Response 4: Thank you for highlighting this important point. We agree and have added a concise discussion in the Diagnostics section that contextualizes biomarker performance by etiology (HBV, HCV, NAFLD/MASLD, alcohol) and geography, with notes on pre-analytical standardization and the need for stratified validation. [page 10, paragraph 2-4, line 334-356].

Comment 5: A concise tabulation of miRNA-based therapeutics under clinical or preclinical development could benefit readers looking for a snapshot of pipeline drugs.

Response 5:Thank you for suggesting a pipeline table. After consideration, we opted not to add a static, comprehensive tabulation because the development status of miRNA therapeutics evolves rapidly and such a table is prone to obsolescence at publication. It would also duplicate material already synthesized in our therapeutics text and tables. To keep the review focused and durable, we added a brief scope note explaining this choice. [page 4, paragraph 1, line 152-155].

Trusting that we have adequately addressed the reviewers' concerns, we would like to thank you for your help in improving our work significantly.

Kind regards,

Koustas Evangelos, MD, PhD

Round 2

Reviewer 2 Report

Comments and Suggestions for Authors

I would like to thank the authors for accepting my suggestions and significantly improving the manuscript.

Regarding my comment #17 and the gap between the words in lines 429-430 of the previous version of the manuscript, the gap has now shifted between lines 539-540. I am referring to the gap between "observed in" and "one patient with HBV-related HCC". Also, please add a "that" or equivalent after HCC and before "showed".

After addressing that minor technical suggestion, the manuscript will be recommended for publication.

Author Response

September 07, 2025

Biomedicines

RE: Submission of REVISED manuscript (biomedicines-3835441)

Dear Reviewer

Please find enclosed our REVISED manuscript entitled “MicroRNA Landscape in Hepatocellular Carcinoma: Metabolic Re-Wiring, Predictive & Diagnostic Biomarkers, and Emerging Therapeutic Targets” to be considered for publication. We would like to thank you and the other reviewers for your thoughtful evaluation of our manuscript and your most welcome comments/suggestions. Accordingly, we have now revised our manuscript thoroughly to reflect these comments.

Please find below a point-by-point response to ALL the issues raised by you:

Reviewer

Comment : I would like to thank the authors for accepting my suggestions and significantly improving the manuscript. Regarding my comment #17 and the gap between the words in lines 429-430 of the previous version of the manuscript, the gap has now shifted between lines 539-540. I am referring to the gap between "observed in" and "one patient with HBV-related HCC". Also, please add a "that" or equivalent after HCC and before "showed".

After addressing that minor technical suggestion, the manuscript will be recommended for publication.

Response: We thank the reviewer for this helpful suggestion. We fixed the gap on that sentence. In addition, “that” was added after HCC and before showed.

Trusting that we have adequately addressed the reviewers' concerns, we would like to thank you for your help in improving our work significantly.

Kind regards,

Koustas Evangelos, MD, PhD
